# Force–reversible chemical reaction at ambient temperature for designing toughened dynamic covalent polymer networks

Mengqi Du [1], Hannes A. Houck [2], Qiang Yin[3], Yewei Xu[1], Ying Huang[1], Yang Lan[4], Li Yang [1,5✉], Filip E. Du Prez [2✉] & Guanjun Chang [1,5✉]

Force-reversible C–N bonds, resulting from the click chemistry reaction between triazolinedione (TAD) and indole derivatives, offer exciting opportunities for molecular-level engineering to design materials that respond to mechanical loads. Here, we displayed that TAD-indole adducts, acting as crosslink points in dry-state covalently crosslinked polymers, enable materials to display reversible stress-responsiveness in real time already at ambient temperature. Whereas the exergonic TAD-indole reaction results in the formation of bench-stable adducts, they were shown to dissociate at ambient temperature when embedded in a polymer network and subjected to a stretching force to recover the original products. Moreover, the nascent TAD moiety can spontaneously and immediately be recombined after dissociation with an indole reaction partners at ambient temperature, thus allowing for the adjustment of the polymer segment conformation and the maintenance of the network integrity by force-reversible behaviors. Overall, our strategy represents a general method to create toughened covalently crosslinked polymer materials with simultaneous enhancement of mechanical strength and ductility, which is quite challenging to achieve by conventional chemical methods.

[1] State Key Laboratory of Environment-friendly Energy Materials & School of Materials and Chemistry, Southwest University of Science and Technology, Mianyang 621010, P. R. China. [2] Polymer Chemistry Research Group, Centre of Macromolecular Chemistry (CMaC), Department of Organic and Macromolecular Chemistry, Ghent University, Krijgslaan 281 S4-bis, B-9000 Ghent, Belgium. [3] Research Center of Laser Fusion, China Academy of Engineering Physics, Mianyang 621900, P. R. China. [4] Melville Laboratory for Polymer Synthesis, Department of Chemistry, University of Cambridge, Cambridge CB2 1EW, UK. [5] Department of Chemical and Biomolecular Engineering, University of Pennsylvania, Philadelphia, PA 19104, USA. ✉email: yanglichem628@126.com; filip.duprez@ugent.be; gjchang@mail.ustc.edu.cn

D ue to their excellent mechanical strength and thermal
stability, polymer materials fabricated with covalent
crosslinks have been widely used in numerous fields of
daily life[1–4]. Most of the covalently crosslinked polymer materials
are employed under ambient conditions, where the covalent
polymer network integrity directly affects polymer materials'
properties and lifetime[5–8]. However, long-term force perturbation
is inevitable in crosslinked polymers and typically induces irre-
versible covalent bond scission, resulting in a chemically damaged
crosslinked network[9]. This phenomenon significantly weakens
the mechanical and functional properties of covalently cross-
linked polymer materials, as well as shorten their service life, and
even bring safety risks to their application.

Generally, a weak force-activated covalent bond can break
preferentially in order to dissipate energy from external force
perturbation[10–12]. Currently, a series of weak force-activated
covalent bonds have been introduced as crosslink points into
covalent polymer networks to increase simultaneously the
mechanical strength and ductility of a polymeric material[9,13,14].
Despite the fact that weak force-activated covalent bonds could be
re-formed by external stimulation[15–18], such as UV or visible
light irradiation, heating or a catalyst, the broken bonds cannot be
reformed in real time under ambient conditions, thereby leading
to irreversible damage within the polymer network in the long-
term application process. In addition, homolytic cleavage into
relatively stable organic radicals showed promising reversible
dissociation/association[19,20]. However, radical recombination
reactions are often susceptible to undergo irreversible reactions,
for example with molecular oxygen, moisture, and other sur-
rounding molecules, which is a concern in the context of dynamic
covalent chemistry[20]. In particular when exposed to ambient
conditions for a long time, radical species are expected to lose
their activity to de-bond and re-bond. Nonetheless, Otsuka
et al.[21] developed a notable example based on a difluor-
enylsuccinonitrile (DFSN) linker whose central C-C bond is
readily cleaved under mechanical stress to generate a relatively
stable radical species. However, the delayed recovery of the DFSN
linkers did not allow for their self-reassociation in real time. As a
result, developing an ambient-stable dynamic covalent bond with
reversible stress-responsiveness in real time would afford a
useful route toward designing multiple force-responsive functions
for covalently crosslinked polymers, such as simultaneously

enhancing mechanical strength and ductility, adaptability to
dynamic environments and network autonomy.

Our strategy to introduce force-reversible C-N bond exchanges
at ambient temperature is inspired by reversible click chemistry
between triazolinedione (TAD) and indole-based building blocks
(Fig. 1a)[22–24]. It is known that the TAD-indole chemistry plat-
form offers a range of selective and predictable covalent links that
are quantitatively formed under equimolar conditions at ambient
temperature without the need for a catalyst, while the corre-
sponding adducts are known to be bench-stable, yet are reversible
upon heating. Indeed, TAD-indole covalent bond exchanges at
elevated temperatures have been well-studied in the context of
covalent adaptable networks[24–26]. Intriguingly, however, the
generated C-N bond in TAD-indole adducts is calculated to be
weaker than conventional C-C or C-N bonds (124 kJ mol$^{-1}$
versus ~350 kJ mol$^{-1}$). This implies that, when embedded within a
conventional polymer network, the C-N bond in TAD-indole
adducts could break preferentially, resulting in efficient energy
dissipation of the mechanical stress. Moreover, the nascent TAD
moieties that are bench-stable have high reactivity to indole
groups and the TAD-indole adducts can rapidly re-form[27]. Based
on this, we hypothesized and investigated, for the first time, the
real-time breaking and re-forming of the force-reversible C-N
bond in TAD-indole adducts to occur at ambient temperature in
dry-state covalently crosslinked polymer materials (Fig. 1b). The
quick, reversible stress-responsiveness could result in polymer
materials with unprecedented enhancement of mechanical
strength and ductility via polymer segment conformational
adjustment as a function of time, thereby maintaining the net-
work integrity.

## Results

**Synthesis of C-N crosslinked polymer materials**. To demon-
strate our conceptual approach toward force-reversible C-N
crosslinked polymer networks, both a linear poly(methyl
methacrylate) (LPMMA, $T_g = 91$ °C) and poly(methyl acrylate)
(LPMA, $T_g = 8$°C) with 2-phenylindole in the side chains have
been synthesized, and subsequently treated with a triazolinedione
crosslinker [4,4'-(4,4'-diphenylmethylene)-bis-(1,2,4-triazoline-
3,5-dione) (MDI-TAD)] to form C-N crosslinked polymer
materials (CPMMA and CPMA, respectively, see Fig. 1c). The

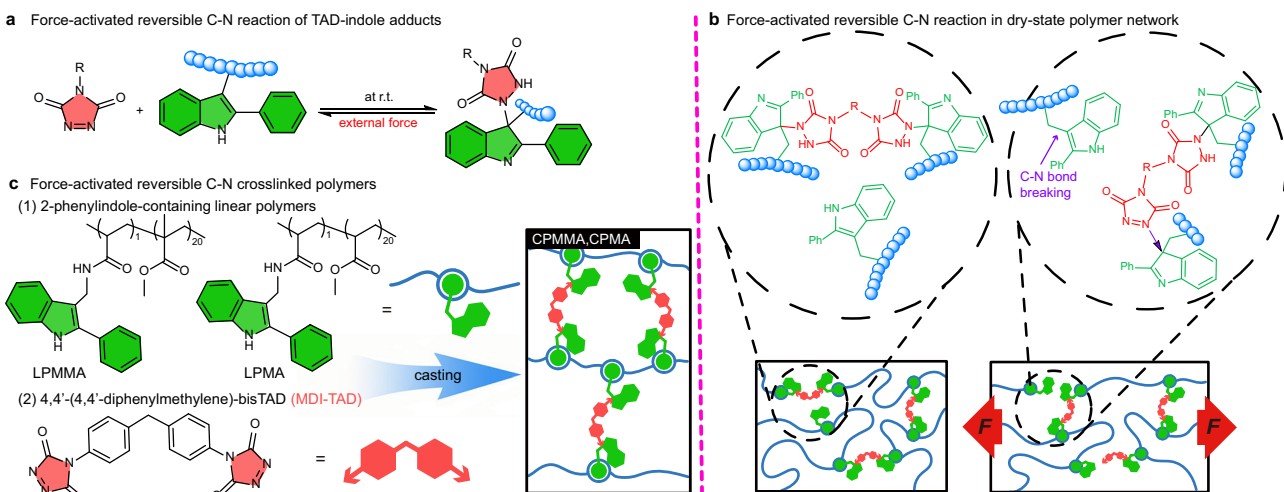

**Fig. 1 Monomer and polymer force-reversible TAD-indole chemistry. a** Scheme of TAD-indole reversible click reactions at ambient temperature, including
a common click reaction and force-reversible reaction. **b** The breaking and reforming of the force-reversible C-N bond occurs sequentially at ambient
temperature in dry-state covalently crosslinked polymers. **c** TAD-based crosslinker (MDI-TAD) and two different linear indole-containing polymers used in
this work to design force-reversible C-N crosslinked amorphous poly(methyl acrylate) (CPMA) and poly(methyl methacrylate) (CPMMA) based materials.

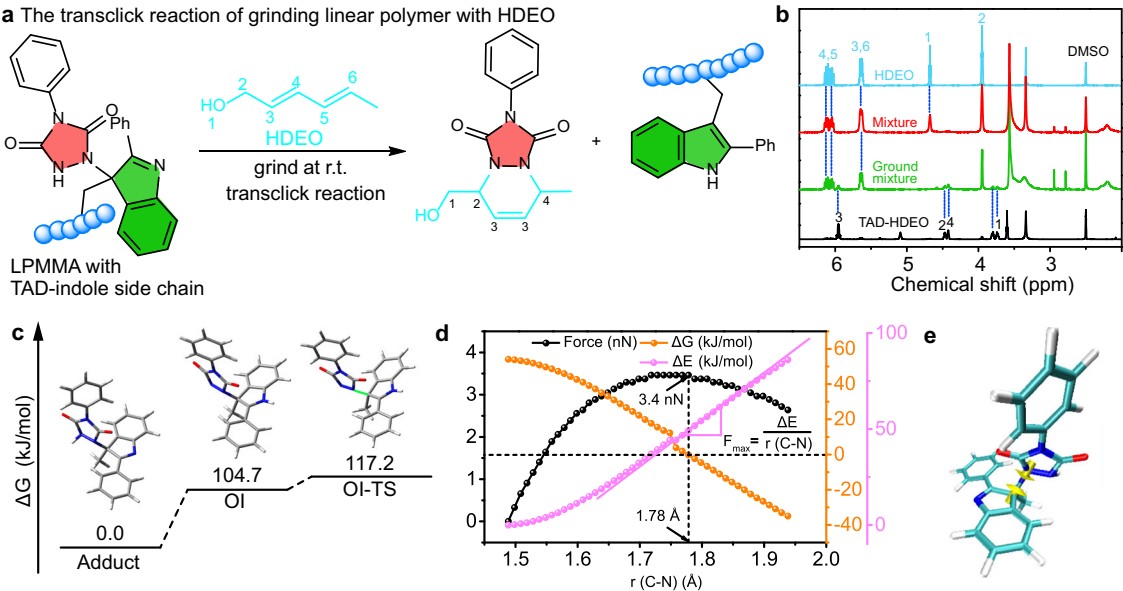

**Fig. 2 Verified force-reversible C-N bond breaking in the dry-state polymer network. a** Transclick reaction of TAD-indole side chain in LPMMA to an irreversible TAD-HDEO adduct, triggered by the external force at ambient temperature. **b** $^1$H NMR spectrum of a small molecule HDEO (blue), the TAD-HDEO reference adduct (black), the mixture of HDEO and LPMMA with TAD-indole side chains (red) and the ground mixture (green) dissolved in DMSO-$d_6$. **c** Free energy profiles (kJ/mol) for TAD-indole adduct, open zwitterionic intermediate (OI) and transition state (OI-TS). **d** Theoretically calculated restored force, free energy and potential energy of dissociation of TAD-indole adducts as a function of C-N bond length at 1 atm, 298.15 K. **e** TAD-indole adduct with yellow arrows along the C-N bond representing the loading direction of the C and N atom when external force is applied in the quantum-chemical calculations.

corresponding polymer films have been prepared by a solvent casting method (see Methods, Supplementary Table 1 and Supplementary Fig. 1–9 for detailed experimental procedure and characterization). High TAD conversion in indole-based polymer networks was confirmed by Fourier-transform infrared spectroscopy (Supplementary Fig. 8). For this study, we selected a polymer containing 5 mol% $N$-((2-phenyl-1$H$-indol-3-yl)methyl) acrylamide (NPI) repeating units as an example to present our results. At ambient temperature, the TAD-indole crosslinked films are semi-transparent, non-adhesive and insoluble solids (see Supplementary Table 2 for soluble fractions). According to the differential scanning calorimetry (DSC) curves, the CPMMA can be considered as a typical hard thermoset ($T_g$ = 120ºC), and CPMA as a soft material ($T_g$ = 18ºC, Supplementary Fig. 10).

**Force-reversible C-N bond breaking**. We performed a series of experiments and simulations to study the molecular mechanism of the microscopic changes in response to alternative external force treatments, including TAD-indole bond breaking, reformation and successive reversible bonding. First, to verify the breaking of the C-N bond in the TAD-indole adduct in the dry state when subjected to an external force, a grinding test was conducted as a method for applying external force on a polymer network[28]. Since TAD-indole adducts can be 'transclicked' upon heating in the presence of a conjugated diene, i.e., 2,4-hexadien-1-ol (HDEO), to generate an irreversible TAD-HDEO adduct (Fig. 2a)[24], the LPMMA film with TAD-indole side chain adducts was cut by scissors and soaked into a 1 M ethanol solution of HDEO. The LPMMA film containing HDEO was received after drying the swollen network at 40ºC for 12 h to form the mixture containing LPMMA and HDEO. The grinding time was 30 min with intervals of 10 min to avoid overheating caused by friction. It was found that, after grinding the mixture, TAD-HDEO adducts were obtained as verified by $^1$H NMR spectroscopy of the dissolved ground powder in DMSO-$d_6$ (Fig. 2b). For comparison,

there is no characteristic peak of the TAD-HDEO in $^1$H NMR for the unground reference sample (Fig. 2b). Thus, the nascent TAD moiety due to mechanical activation during the applied grinding force is believed to be consumed in the presence of HDEO.

Furthermore, our calculations indicated that pulling C-N linkages of the TAD-indole adducts promotes their dissociation (see Methods)[29–31]. The $\Delta G$ dependence of the constrained TAD-indole adducts on the force resulted from the differences in stabilization of the ground states and the transition state (Fig. 2c, Supplementary Table 3). As excepted, with the increasing force, the de-stabilizing TAD-indole adducts lead to the reaction equilibrium to favor dissociation, with a concomitant decrease in the $\Delta G$ (Fig. 2d)[11]. Applied external force on the polymers is capable of activating internal C-N bonds of the TAD-indole adducts and reducing the apparent activation barrier for the dissociation reaction, implying that free states of TAD and indole groups are more favorable with higher stress (Supplementary Fig. 11). This result is consistent with prior reports that applied force affects the reversible reaction tendency by decreasing the activation energy barrier for the dissociation reaction[32,33]. Yellow arrows in Fig. 2e show the loading direction along the C-N bond of the stressed TAD-indole ensembles. In the dissociation of TAD-indole adducts pulled at the C-N linkages, such changes occur at a maximum restored force of 3.4 nN, which is a comparable value to that of previously reported weak force-activated covalent bonds[11,34–36]. Thus, our theoretical calculations indicated that the C-N bonds in TAD-indole adducts could break preferentially as compared to more traditional chemical covalent bonds.

**Ambient force-activated reversible behavior**. The force-reversible behavior was further investigated via mechanical experiments. Since the TAD azo-bond (-N=N-) is typically consumed upon reaction, the corresponding n-π* absorption peak, giving rise to its characteristic red color (Supplementary Fig. 12),

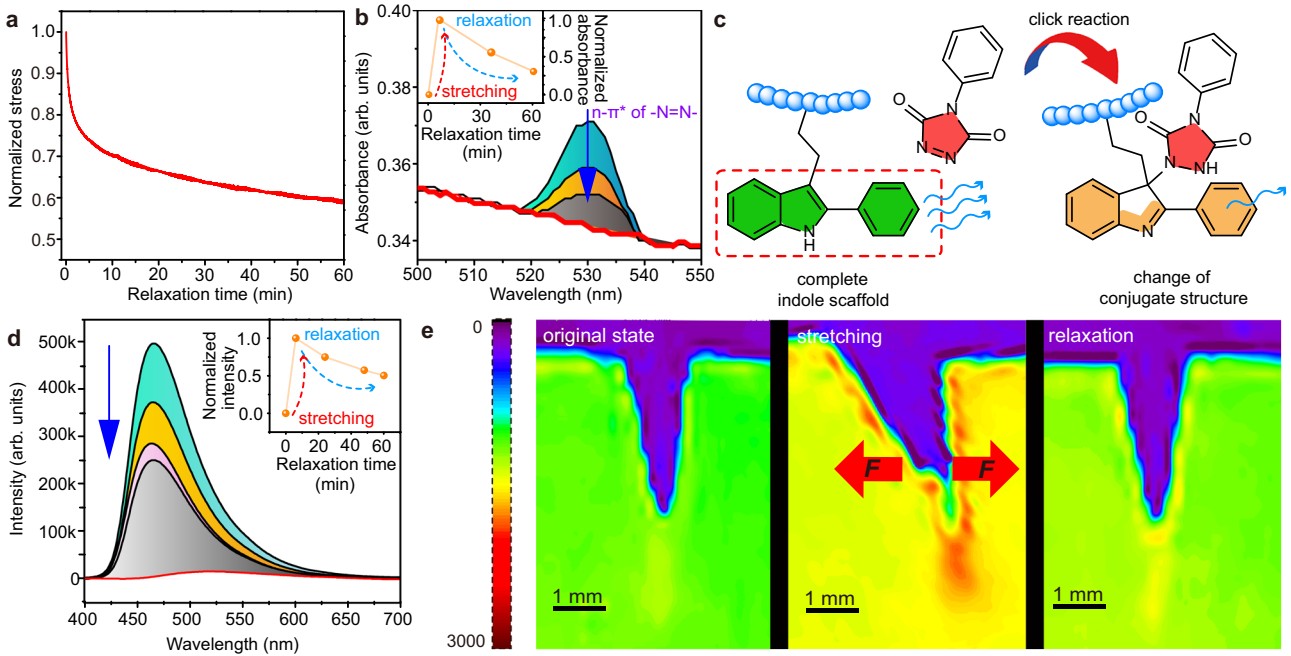

**Fig. 3 Force-reversible behavior of the C-N bond in dry-state polymer film at ambient temperature. a** Stress relaxation curve of CPMMA at ambient temperature detected by universal tensile test machine. **b** The in situ recorded stress relaxation–UV/vis spectra and the UV/vis spectrum of the original film (red line). **c** Representation of fluorescence intensity reduction upon changing the conjugated structure of the indole scaffold after TAD-indole click reaction (colors refer to the electron cloud, cf. Supplementary Fig. 18). Blue wave arrows indicate fluorescence intensity. **d** The in situ stress relaxation–RF spectra for CPMMA film was used for tracking of the fluorescence intensity changes at ~460 nm. **e** Fluorescence mapping where bonds break and re-form in the CPMMA film during one force cycle.

disappears upon the formation of the TAD-indole adduct[22,37]. In situ stress relaxation–UV/vis spectra were carried out on a stretched spin-coated CPMMA film by using a simple relaxation fixture device (Supplementary Fig. 13, 14). The results clearly showed the -N=N- absorption peak arising upon applying stress to the CPMMA film (Fig. 3a, b), indicative for in situ TAD release. The peak intensity gradually decreases over time during stress relaxation, suggesting C-N bond reformation of TAD-indole moieties within the network (Fig. 3b). A similar phenomenon was observed for the lower $T_g$ CPMA film at ambient temperature (Supplementary Fig. 15).

In addition to the response in UV/vis, we found that the linear LPMMA film exhibits strong fluorescence ($\lambda_{exc} = 365$ nm), resulting from the photoluminescence properties of the indole units[38]. After curing, the fluorescence intensity of the crosslinked CPMMA film decreased significantly (Supplementary Fig. 16, 17). Such fluorescence intensity decrease was also observed for crosslinked CPMA compared to linear LPMA, due to the change in the conjugated structure of the indole scaffold before and after TAD-crosslinking (Fig. 3c, Supplementary Fig. 18)[39]. The LPMMA films showed excellent photostability under 365 nm when irradiated for one hour (Supplementary Fig. 19). Based on this, the in situ stress relaxation–RF spectra were used for tracking fluorescence intensity changes for CPMMA and CPMA films during the process of mechanical relaxation. As expected, the results showed that the fluorescence intensity initially increased upon elongating the films, but decreased during stress relaxation to 50% after ca. 60 min and 5 min for CPMMA and CPMA films, respectively (Fig. 3d and Supplementary Fig. 20). In addition, the fluorescence intensity map of CPMA and CPMMA on a force cycle was used to establish a direct pixel-by-pixel qualitative visualization of the force-reversible C-N bond breaking and reforming in dry-state polymers at ambient temperature (Fig. 3e and Supplementary Fig. 21). The essential step[10,40,41] was

to apply one force cycle on a notched polymer sample to obtain an intensity-colored map of the mechano-reversible C-N bond behavior in TAD-indole adducts, including breaking and reforming. We observed that the force-reversible C-N bond breaking is very localized in front of the crack tip, exhibiting enhanced fluorescence. The fluorescence intensity decreased significantly when removing the external force, which can be ascribed to the real-time conversion of nascent TAD and indole moieties with the change of stress. The change in fluorescence, combined with the UV/vis response, thus supports the reformation of the C-N bond of the TAD-indole adduct at ambient temperature when no external force is applied.

**Mechanical properties**. The crosslinking density of the networks was readily controlled by the monomer feed ratios of TAD-indole adduct to polymer chain repeat unit. The resulting thermosets were denoted as CPMMA-x% and CPMA-x%, where x was the molar fraction of TAD-indole adduct in the feed (varied between 2.5 and 10 mol%). Uniaxial tensile tests showed that the mechanical properties initially increased and then decreased with increasing crosslinking density (Fig. 4a, b), with optimal mechanical properties observed for CPMA-5% and CPMMA-5%. Compared to the corresponding linear polymer films, for CPMA-5% as a soft material, the dynamic net effect is an extraordinarily high 6-fold increase in tensile strength (from $\sigma_{LPMA} = 3.5$ MPa to $\sigma_{CPMA} = 22.1$ MPa), an 8-fold increase in the yield stress (from 0.7 MPa to 5.6 MPa) and a corresponding 2.2-fold increase in elongation at break (from 280% to 610%). Accordingly, there is a 15-fold increase in tensile toughness (energy density from 4.8 MJ cm$^{-3}$ to 74.1 MJ cm$^{-3}$). For the hard material CPMMA-5%, a 2-fold increase in tensile strength ($\sigma_{LPMMA} = 62.2$ MPa vs. $\sigma_{CPMMA} = 114.9$ MPa), a 2-fold increase in elongation at break (from 22% to 44%) and a 3-fold increase in tensile toughness (from 12.5 MJ cm$^{-3}$ to 34.7 MJ cm$^{-3}$) was observed. In contrast to

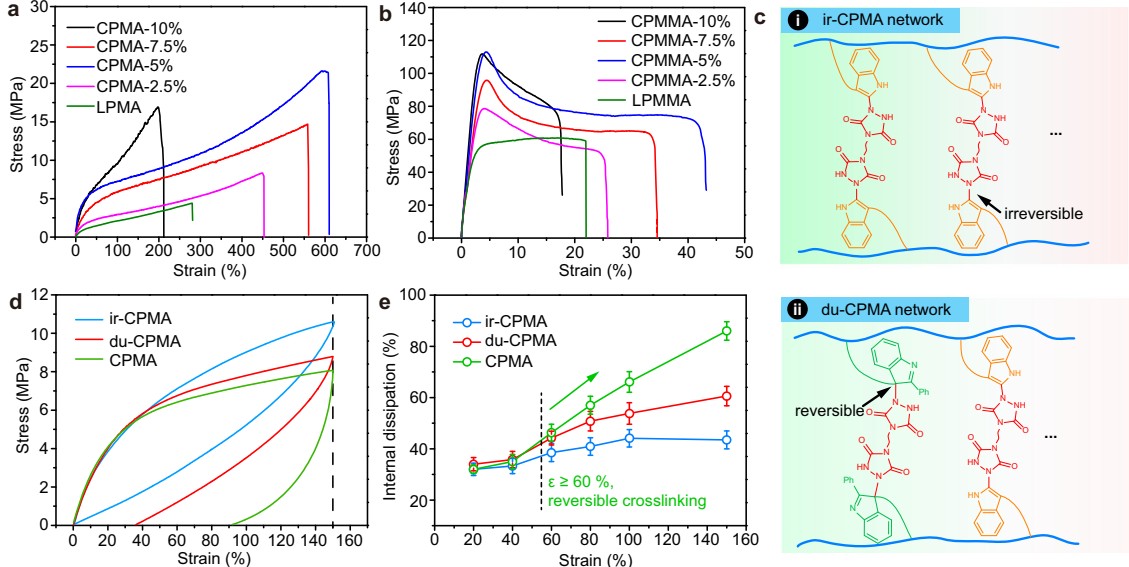

**Fig. 4 Uniaxial tensile and recovery.** Stress as a function of strain of **a** CPMA ($\dot{\varepsilon} = 20$ mm min$^{-1}$) and **b** CPMMA ($\dot{\varepsilon} = 2$ mm min$^{-1}$) films. **c** Structural representation of irreversible crosslinked PMA (ir-CPMA), and dual crosslinked CPMA (du-CPMA) containing a 1:1 ratio of reversible and irreversible crosslinks. **d** One cyclic stretching of different crosslinked films at 40ºC. **e** The mechanical internal friction dissipation of the three networks at different strains. The error bar was derived from five tests of internal dissipation.

traditional thermosets[42–45], such as epoxy resin and unsaturated polyester resins, the resulting CPMMA-5% exhibited simultaneous enhancement of mechanical strength and ductility.

To evaluate the effect of the addition of the TAD crosslinker on the polymeric mechanical properties, a foreseen control experiment was performed by the addition of Ph-TAD, the monofunctional TAD modification agent that has a similar chemical structure as the crosslink points. Thus, a small amount of Ph-TAD (5 mol%), was introduced into the LPMA or LPMMA polymer materials (LPMA* and LPMMA*). Uniaxial tensile tests showed that the introduction of a small amount of Ph-TAD has no significant effect on the mechanical properties (Supplementary Fig. 22). Therefore, the effect of the addition of a small amount of TAD crosslinker itself, acting as a plasticizer or anti-plasticizer, on the mechanical properties was not considered further in this study.

To investigate the energy dissipation behavior of the force-reversible C-N crosslinked polymers, a series of control experiments were performed. For this, we fabricated three types of polymer films with similar crosslinking densities: a first network consisting of the reversible TAD-indole crosslinked CPMA (CPMA, cf., Fig. 1b), a second network based on TAD-indole crosslinks without the indole 2-phenyl substituent that give adducts that are considered irreversible (ir-CPMA, see Fig. 4c and Supplementary Fig. 23, 24), and a third dual crosslinked CPMA network (du-CPMA) containing a 50:50 ratio of reversible and irreversible TAD-indole crosslinking points (Fig. 4c, Supplementary Table 1)[23]. The reversible vs. irreversible linkage of TADs with indoles highly depends on the substituents in the indole C2 and C3-position. When no substituent on the indole C2-position is present, as is the case for NIAM, TAD-addition will result in the thermodynamically favored adduct in the C2-position (Supplementary Fig. 23). Consequently, the retention of aromaticity in the C2-indole adduct renders the TAD addition irreversible in practice[46]. It increased the bonding energy of the C-N bonds in TAD-indole adducts which is comparable with that of the traditional covalent bonds (361.3 kJ/mol vs ~350 kJ/mol, Supplementary Fig. 24). The cyclic stretching tests were performed at 40ºC in order to maintain the same physical state for all materials

(rubber, Supplementary Fig. 25). Residual strain after one cyclic stretching depended on the type of crosslinking modes (Fig. 4d). Increased amounts of reversible TAD-indole adducts were conducive to reduce molecular internal stress due to the continuous reversible behavior embedded by the C-N bond of the TAD-indole adducts, thus resulting in the lowest strain recovery ($\varepsilon_{recovery} = 58$ %) for CPMA compared to that of du-CPMA ($\varepsilon_{recovery} = 114$ %) and ir-CPMA ($\varepsilon_{recovery} = 148$ %).

To characterize potential changes in the mechanical properties across different bonding motifs, a quantitative assessment by means of Eq. (1) was used to reveal the relationship between the energy of loading and dissipation in the hysteresis:

$$\psi = \frac{U_{dis}}{U_{loading}} \times 100\% \tag{1}$$

where $\psi$, $U_{loading}$ and $U_{dis}$ refers to the mechanical internal friction dissipation and the energy of loading and dissipated energy, respectively. We first tested one cyclic stretching of three crosslinked polymer films at different strains (20%, 40%, 60%, 80%, 100% and 150 %, Supplementary Fig. 26 and Fig. 4d). At small applied strains ($\varepsilon \leq 40$ %), three covalent crosslinked polymer samples had similar $U_{dis}$, $U_{loading}$ and $\psi$ values (Fig. 4e), indicating that the external force at low stain is too low to break the reversible C-N bonds, making these to act as fixed crosslink points. At higher applied strains, i.e., $\varepsilon \geq 60$ %, continuous dissociation and reconstruction of the C-N bonds in the TAD-indole adducts resulted in a large increase of $U_{dis}$ and $\psi$ for the CPMA samples, as compared with the ir-CPMA and du-CPMA samples (Fig. 4e). This implies that the force-reversible C-N bonds can efficiently dissipate the energy caused by external forces.

**Recovery mechanism and kinetics.** Measurements for load, time and fluorescence intensity were captured simultaneously during cyclic tensile testing using a combination of an optical and mechanical setup. A photo and schematic of the setup are shown in Fig. 5a. The force-reversible behavior of TAD-indole adducts during the stretching tests is manifested as the thickness-corrected normalized fluorescence intensity value ($I_{fl,norm}$, details in Methods). At

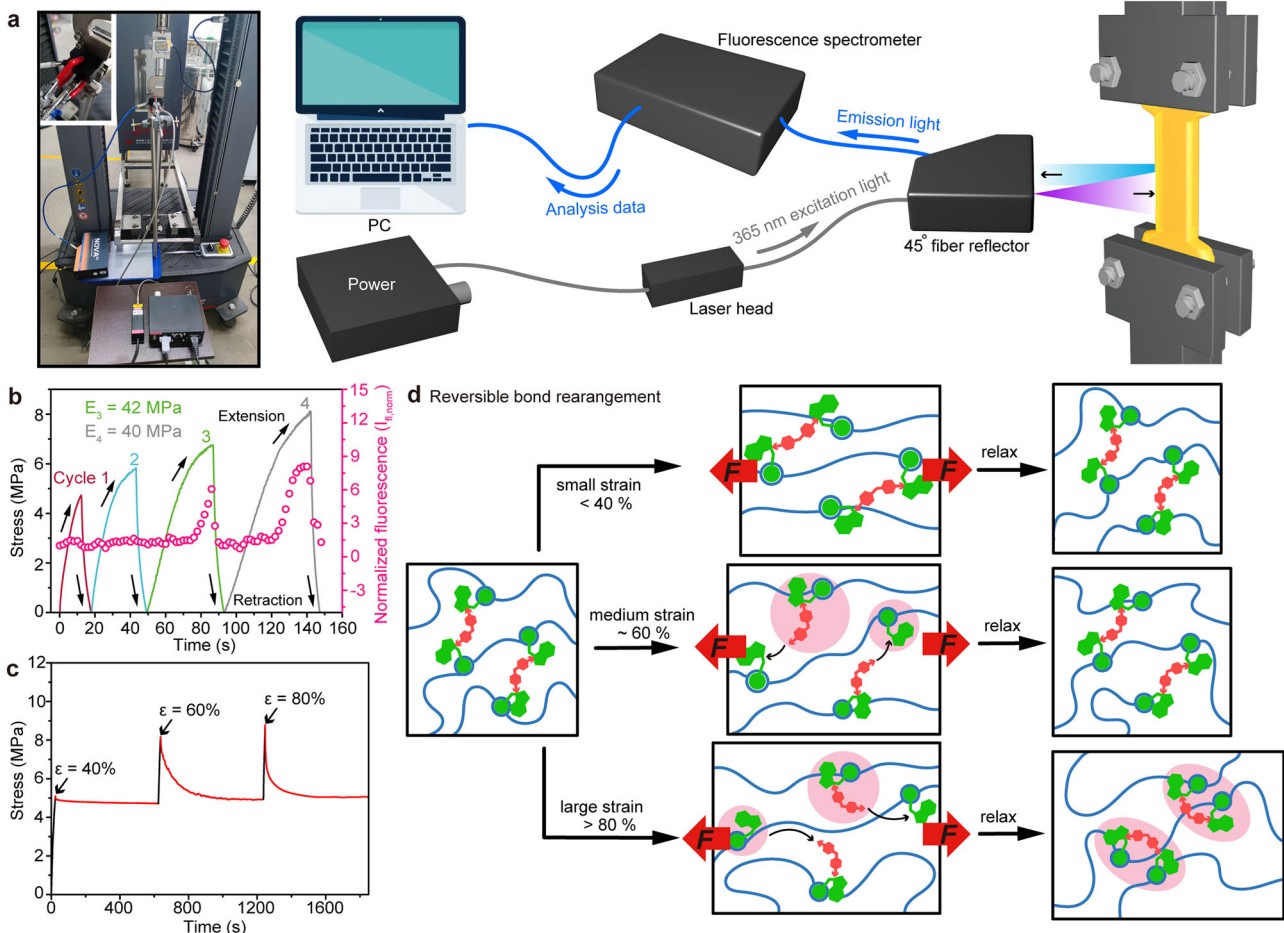

**Fig. 5 The recovery mechanism and kinetics of the reversible C-N bonds. a** Design of the customized setup, combining the use of a universal testing machine and portable fluorescence spectrometer, $\lambda_{exc} = 365$ nm. **b** Simultaneously obtained stress and thickness-corrected normalized fluorescence intensity as a function of strain. **c** The repeated stress relaxation of CPMA at $\varepsilon = 40\%$, 60% and 80%, respectively. Relaxation time is 10 min for every stage. **d** Schematic representation of proposed bond rearrangements for the strain recovery.

small applied strains ($\varepsilon \leq 40$ %, Fig. 5d), $I_{fl,norm}$ almost did not change (cycle 1 and 2 in Fig. 5b), and the unbroken reversible C-N bonds act as conventional chemical crosslinks. The resulting macroscopic deformation corresponds to chain conformation changes, and the entropy of the system is reduced. The unbroken bonds preserve shape-memory and drives the network entropically back to the unstretched state after the force is released (Supplementary Fig. 26a, b). At medium applied strains ($\varepsilon$, ~60 %), the force-reversible C-N bonds in the network can be assumed to begin to break (Fig. 5d), thus causing $I_{fl,norm}$ to sharply increase (cycle 3 in Fig. 5b) with a large deformation recovery of the network (Supplementary Fig. 26c). Finally, at larger applied strains ($\varepsilon \geq 80$ %), the breakage of reversible C-N bonds occurs during elongation while their reformation can happen at newly accessible sites (Fig. 5d). Thus, no entropy is gained during the macroscopic deformation leading to network topological rearrangement. In essence, the network is in this case permanently deformed (non-recoverable) from the standpoint of thermodynamics, and the residual strain would be observed (illustrated in Fig. 4d and Supplementary Fig. 26d, e)[47]. In addition, the cleavage and reconstruction of TAD-indole adducts reaches a dynamic equilibrium (fixed $I_{fl,norm}$), revealing the same reaction rate of force-reversible C-N de-bonding and re-bonding at high strain (cycle 4 in Fig. 5b). During retraction, the fluorescence intensity originating from the recovery of TAD-indole adducts decreased rapidly in real time with the stress being removed (cycle 3 and 4). The modulus of the polymer network after TAD-indole

adducts re-association maintained around 40 MPa (cycle 3 and 4). Breaking and reforming kinetics of the dynamic bonds in the network will significantly affect the mechanical properties of the polymer materials. A repeated stress relaxation curve corroborates the fast kinetics of the force-reversible C-N bond in the network ($\varepsilon > 60\%$, Fig. 5c), which is essential for the design of tough, high-performance polymer materials.

In this study, we have combined the force-reversible C-N bond in TAD-indole adducts, which displays reversible stress-responsiveness in real time already at ambient temperature, in both high $T_g$ and low $T_g$ covalently crosslinked polymer materials. This approach endows the covalently crosslinked polymers with an unprecedented enhancement of mechanical strength and ductility. The key innovation of our approach is thus the use of a building block for the construction of force-activatable crosslinks that can be installed in the solid state without any external stimulus than ambient temperature. Such ambient force-reversible C-N covalent crosslinking could thus be considered as a new molecular platform for designing high-performance polymer materials.

## Methods

**Synthesis of *N*-(1*H*-2-phenyl-indole-3-methyl) acrylamide, NPI.** To a stirred solution of 2-phenylindole (19.32 g, 0.1 mol) and *N*-methylolacrylamide (12.13 g, 0.12 mol) in dichloromethane (200 mL) was added, dropwise, a suspension of anhydrous aluminum chloride (1.34 g, 10 mmol) in dichloromethane (20 mL) in an ice bath. The mixture was gradually warmed to room temperature after 2 h of

stirring in an ice bath. After another 48 h of stirring at room temperature, the reaction mixture was added to the ice dilute sulfuric acid solution (the molar ratio of sulfuric acid to aluminum trichloride is 3:2). DCM was used to extract for many times. The organic phase was collected and washed with deionized water, dried using $Na_2SO_4$, and concentrated in vacuo to obtain the crude residue. The latter was purified by silica gel column chromatography using petroleum ether/ethyl acetate = 2:1 ($R_F$ = 0.75 with TLC eluent petroleum ether:ethyl acetate = 1:2) to afford the indole derivative (NPI, Supplementary Fig. 1, 2, yield: 73%). $^1$H NMR (600 MHz, DMSO-$d_6$): δ = 11.373 (s, N$H$), 8.412 (s, N$H$), 7.701 (s, Ar$H$), 7.598 (s, Ar$H$), 7.529 (t, Ar$H$), 7.404 (m Ar$H$), 7.137 (t, Ar$H$), 7.031 (t, Ar$H$), 6.291 (m, C$H$), 6.178 (d, C$H_2$), 5.591 (d, C$H_2$), 4.509 (d, C$H_2$). $^{13}$CNMR (125 MHz, DMSO-$d_6$): δ (ppm) = 164.853 (C), 136.414 (C), 132.634 (C), 132.223 (C), 129.249 (C), 128.830 (CH), 128.600 (CH), 128.228 (CH$_2$), 125.601 (C), 122.321 (CH), 119.596 (CH), 119.416 (CH), 111.703 (C), 108.761 (CH), 33.944 (CH$_2$). HRMS (m/z): *calc.*: 277.1331 [MH]$^+$, *found*: 277.1296 [MH]$^+$.

**Synthesis of N-(1H-indole-3-methyl) acrylamide, NIAM.** Indole (5.85 g, 0.05 mol), N-methacrylamide (7.2 g, 0.07 mol) and absolute ethanol (100 mL) were added sequentially in a 250 cm$^3$ Erlenmeyer flask, and wait until the N-methacrylamide is completely dissolved, anhydrous aluminum trichloride (4.0 g) was slowly added. Then the reaction was stirred in a water bath at 25 °C for 3 days. During this period, the system gradually changed from colorless to red. Most of the ethanol is removed by rotary evaporation, and the concentrated solution is poured into a dilute sulfuric acid solution (the molar ratio of sulfuric acid to aluminum trichloride is 3:2) and stirred to remove the catalyst aluminum trichloride. After suction filtration and drying, a brick-red crude product is obtained, which was purified by means of column chromatography using petroleum ether/ethyl acetate = 2:1 as eluent ($R_F$ = 0.45 with TLC eluent petroleum ether:ethyl acetate = 1:2) to obtain pure N-(1H-indole-3-methyl) acrylamide (NIAM, Supplementary Fig. 3, 4, yield: 65.5 %). $^1$H NMR (600 MHz, DMSO-$d_6$): δ = 10.930 (s, N$H$), 8.331 (s, N$H$), 7.549 (d, Ar$H$), 7.367 (d, Ar$H$), 7.272 (d, Ar$H$), 7.082 (d, Ar$H$), 6.984 (d, Ar$H$), 6.249 (m, -C$H$=CH$_2$), 6.139 (m, -CH=C$H_2$), 5.572 (d, -CH=C$H_2$), 4.486 (d, Ar-C$H_2$-). $^{13}$CNMR (125 MHz, DMSO-$d_6$): δ (ppm) = 164.648 (C), 136.497 (C), 132.719 (CH), 126.705 (C), 125.198 (C), 124.069 (CH), 122.020 (CH), 119.139 (CH), 119.002 (CH), 112.564 (C), 111.717 (CH), 34.539 (CH$_2$). HRMS (m/z): *calc.*: 201.1024 [MH]$^+$, *found*: 201.0983 [MH]$^+$.

**Synthesis of linear and crosslinked polymers.** To a two-necked flask equipped with a magnetic stirrer, a nitrogen outlet and inlet, MA or MMA (100 mmol), NPI (5 mmol), AIBN (0.015 wt%) and DMF (3 mL) were added. The reaction mixture was evacuated and flushed with high-purity nitrogen. The reaction mixture was heated to 75 °C under a nitrogen atmosphere stirring for 0.5 h. The 0.1 wt% ABVN was added to the prepolymer solution and then was heated at 55 °C for 24 h. The resulting polymer solid finally was heated at 120 °C for 2 h. The final polymer was placed in vacuum dryer at 80 °C and 0.09 MPa for 12 h to drying to afford LPMA and LPMMA. A typical synthetic procedure for polymer CPMA is illustrated as an example. The synthesis of MDI-TAD refers to the reported literature and is shown in Supplementary Fig. 5[24]. LPMA (3 g) was completely dissolved in 1,4-dioxane (30 mL) at 80ºC for about 2 h. LPMMA was dissolved using DMF. The polymer solution was placed in a refrigerator to decrease the temperature (about 15ºC) of the polymer solution. This process slows down the crosslinking speed during mixing. MDI-TAD was dissolved in a small amount of 1,4-dioxane solvent and placed into the refrigerator in advance (DMF was used in CPMMA system). The polymer solution is placed on a stirrer with a high stirring speed (1000 rpm). The MDI-TAD solution was poured into the polymer solution and the resulting red solution was stirred for 20 s. Then, the solution was cast onto a silicon rubber sheet, with the organo-sol transition being observed within half a minute. The solvent was evaporated in a vacuum oven at 80 °C for 12 h to get the targeted polymer film (about 0.2 mm thick).

**Instrumentation.** Uniaxial tensile testing was carried out on electromechanical universal tester machine (MTS, CMT4304) with 50 N sensor at a strain rate of 2 mm/min (for PMMA based materials) or 20 mm/min (for PMA based materials) and a temperature of 25 °C. The 3.5 cm length dog bone-shaped specimens were prepared by cut-off knife followed by precise machining. The stress at break and elongation at break were obtained from the stress-strain experiment with at least three identical specimens and reported as averaged values. The stress relaxation experiments were carried out on electromechanical universal tester machine (MTS, CMT4304). Samples were first loaded uniaxially at a strain rate of 20 mm/min or 2 mm/min to a strain of 80% or 10%. Load and strain are measured with time at constant grip displacement. DSC measurements were carried out on a TA Instruments Q2000 calorimeter. Samples were sealed into a standard aluminum pan. A sealed empty pan was used as reference. The scan was carried out in nitrogen atmosphere at a flow rate of 200 ml/min at a scanning rate of 10 °C /min. Final data were collected for three full cycles which were identical. The inflection points of the DSC traces were used for the determination of the glass transition temperatures, $T_g$.

**In situ stress relaxation–UV/vis absorption spectra.** UV/vis absorption spectra were measured in absorption mode using UV-2600, which was equipped with integrating sphere when testing polymer films. Solid UV/vis absorption spectra were measured by placing a polymer coated quartz sheet in the passageway of incident light using quartz sheet as the background. Both of stretching sample and simple fixture device (Supplementary Fig. 13) was directly placed in the passageway of incident light to test the relaxation–UV/vis absorption spectra in situ. About 20 μm thick films were prepared by spin coating method (Supplementary Fig. 14). Conversion of the transmission-wavenumber to the absorption-wavenumber was carried out using the Beer-Lambert law.

**In situ stress relaxation–RF spectra.** The fluorescence emission spectra, including in situ relaxation–RF spectra, were obtained by using a RF-6000 spectrophotometer. The emission spectra (400–700 nm) were recorded using an excitation wavelength of 365 nm and 5 nm (excitation)/5 nm (emission) slit widths. The stretching sample and the simple fixture device (see Supplementary Fig. 13) was directly placed in the sample pool to test the in situ stress relaxation–RF response.

**Combined mechanical and optical experimental setup.** The experimental setup, shown in Fig. 5a, consists of an uniaxial testing machine with the fluorescence emission recorded by a portable fluorescence spectrometer (NOVA 2 s, ideaoptics, China). This setup allowed for the simultaneous measurement of load-displacement data and fluorescence detection (indicating TAD-indole adducts being activated to dissociate). A 365 nm laser diode was used as the excitation source for fluorescence imaging. The laser irradiates the sample surface through the optical fiber, and the 45 degree optical fiber reflector collects the emitted light for analysis by the portable fluorescence spectrometer. Fluorescence intensity was analyzed in order to obtain relative quantities for activation. A relative measure of fluorescence intensity, $I_{fl,raw}$, was obtained from the portable fluorescence spectrometer. The thickness of the film during the stretching is recorded by a camera. The concentration of the mechanophore decreased during stretching as the sample thinned due to Poisson effects. An adjustment was made to correct for this effect based on measured values for the length stretch ratio λ, from the grip displacement and videotaped width stretch ratio $λ^w$, where $λ^w$ = width/width$_0$[48]. The normalized fluorescence value was divided by the equilibrium value at λ = 1, $I_{fl,raw,0}$:

$$I_{fl,norm} = \frac{I_{fl,raw} \lambda \lambda^w}{I_{fl,raw,0}} \qquad (2)$$

**Quantum-chemical calculations.** All computations were carried out with the Gaussian 16 Revision A.03 software package. The Berny algorithm was used to locate stationary points. Very tight convergence criteria and ultrafine integration grids were used in all optimizations. The calculations of analytical frequencies on converged constrained molecules are valid because the molecule plus its infinitely-compliant constraining potential is a stationary point[49,50]. To start with, the B3LYP/6-31 + G(d,p) level of theory[51] was used for geometry optimizations. A thorough conformational analysis was performed on all reactants and products to identify the most plausible conformers. The B3LYP functional has been proven to produce good geometries but is less accurate for energy calculations. Therefore, energies were refined with the M06-2X functional[52,53], which is able to account for dispersion effects, and a 6–31 + +G(d,p) basis set. Thermal free-energy corrections were taken from B3LYP/6-31 + G(d,p) optimizations at 1 atm and 298.15 K. The above calculation method has been proved its accuracy in TAD system[24]. The enforced geometry optimization (EGO) model was used as the computational method that considers force explicitly under isotensional stretching to study the maximum force of C-N bonds of TAD-indole adducts[29]. The mechanical stretching of a molecule was simulated by relaxed potential energy surface scans. The distance r between the two atoms (C and N atom) on which the force is applied is taken as the reaction coordinate. Starting from the equilibrium geometry, r(C-N) is elongated in fixed small steps (0.05 Å), and with the distance r fixed. An otherwise fully relaxed geometry optimization is performed at each step, giving the potential as a function of r. The maximum slope of potential determines the maximum force of the C-N bonds.

## Data availability

The data that support the findings of this study are available from the authors on request, see author contributions for specific data sets.

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

## Acknowledgements

This work was supported by the National Natural Science Foundation of China (21973076 and 22006122), the Sichuan Talent Fund for Distinguished Young Scholars (2021JDJQ0033), the Applied Basic Research Programs of Sichuan Science and Technology Department (2021YJ0059), the postgraduate Innovation Fund Project by Southwest University of Science and Technology (20ycx0020). Guanjun Chang and Li Yang are grateful for financial support from the China Scholarship Council. Filip Du Prez thanks BOF-UGent and FWO for financial support. We thank the Southwest Computing Center of the China Academy of Physics Engineering for their support of computer simulation.

## Author contributions

F.D.P., G.C., L.Y. and M.D. conceived the research. M.D. carried out the experiments. Q.Y. provided software and technical support. Y.X., Y.H. and Y.L. analyzed the data. F.D.P., G.C. and L.Y. provided research funding. F.D.P., G.C., H.H. and M.D. wrote and revised the paper. All the authors discussed the results and commented on the manuscript at all stages.

## Competing interests

The authors declare no competing interests.
