## [Peer Review File · Nature Communications]

Force–Reversible Chemical Reaction at Ambient Temperature
for Designing Toughened Dynamic Covalent Polymer NetworksReviewers' Comments:

Reviewer #1:

Remarks to the Author:

The manuscript by Du et al reports on the remarkable improvement of mechanical properties of crosslinked polyacrylate networks containing reversible covalent bond between TAD and phenylindole. The manuscript is clear in its message, the experiments are described clearly and support the conclusions. Interesting features are the intrinsic signalling of bond scission by spectroscopic means. The manuscript is somewhat sloppy in its referencing (I found that many references do not correspond to the numbers in the reference list), and English usage is at places insufficient to understand the exact intention of the authors. For example in the caption to Fig. 2, the term 'forced direction' is not clear, and the word 'strengthening' in Fig 3b should probably be 'strengthening', but I am not sure. These weaknesses do not mean that the work should not be published, minor revision is sufficient. I do however feel that the novelty of the work is insufficient for publication in Nature Communications. There are remarkable parallels with the work of Otsuka and coworkers (Sakai et al. Visualization and Quantitative Evaluation of Toughening Polymer Networks by a Sacrificial Dynamic Cross-Linker with Mechanochromic Properties. ACS Macro Lett. 2020, 9 (8), 1108–1113. <https://doi.org/10.1021/acsmacrolett.0c00321>.)

Both studies use a reversible covalent bond to crosslink acrylates and demonstrate both toughening and visualization. Key features such as dissociation energy (124 vs 98 kJ/mol) are similar. Because the chemistry of the reversible bond is different, and the reported toughening is stronger in the current manuscript publication of this work is certainly of interest, but does not have the novelty to merit publication in NC. I recommend publication in a polymer journal after revision that includes a comparison with the paper of Otsuka, and correction of the errors noted above (referencing and language)

Reviewer #2:

Remarks to the Author:

This work addresses a quite interesting topic and is potentially well suited for publication in NCOMMS. The central hypothesis is that introducing a room-temperature forming dynamic covalent bond as cross-linker provides a site of energy dissipation that can also rapidly re-form at room temperature. This could potentially lead to longer-life materials.

The presented evidence is qualitatively supportive of/consistent with the hypothesis, but additional and important control experiments are needed. Changes in bulk properties can arise from very small structural perturbations, and the addition of the TAD cross-linker itself could act as a plasticizer and/or anti-plasticizer that complicates the origins of the changes in mechanical properties. It is possible that while some bond breaking and bond re-forming are occurring, the changes in mechanical properties are due to other effects altogether. So the data in Fig 4a and 4b is difficult to interpret.

The data in Fig 4d/e is more compelling, but the difference in high strain behavior can be greatly influenced by small differences in cross-linking density. I am especially struck by the fact that the green curve eventually stays at higher modulus than the blue curve, which is counter to the picture presented of easier bond breaking in green than blue. One explanation is that the rate of cross-link formation in the irreversible adducts is different from the rate of reversible adducts, leading to "patches" of one and the other. If this is the case, it would also mean that the relative diffusion/reactivity in the two systems is different in forming the thermoset networks, and this could contribute even to the behavior of red vs. green, etc...

Along those lines, one worries that the comparison of Fig. 4e is potentially misleading. Is the same trend in energy dissipation observed at other strains (e.g., 20%, 40%, 60%, etc...)?

The optical data need to be shown in greater detail. There is no "zero" absorbance on the plots, so it is not clear how large the change in intensity relative to final intensity is. In particular, one wonders about the potential for photobleaching (not addressed), and what fraction of the "reacted" TADs are actually forming mechanically active cross-links. How does the modulus of the network after this type of self-healing compare to the initial modulus?

Control experiments with HDEO and no grinding should be performed, as well as grinding of linear polymers with TAD-indole side chain adducts (or small molecule adducts within the cross-linked material) to rule out heat induced dissociation or the presence of trapped, small amounts of initially unreacted TAD.

On page 6, the authors state that the "larger stabilization of ground state" makes the system more reactive, but this seems backwards. Ground state stabilization leads to lower reactivity.

The discussion of "force-activated bonds" is a bit too general, as even conventional covalent bonds are force-activated to break (as noted elsewhere in the paper). This terms should be better defined in the context of the current paper.

References 13 and 14 are used to support the statement that "Currently, a series of force-activated covalent bonds have been introduced as crosslinking points into covalent polymer networks to increase simultaneously the mechanical strength and ductility of a polymeric material." But I do not see discussion of force-activated bonds in this way in what are very general reviews of dynamic covalent bonds in polymers. The authors might be better served by pointing to primary literature here.

The computed force profile in Fig 2e is very strange and uncharacteristic of most COGEF type calculations. The computational details are quite short and more should be provided. But the downward curvature at high forces is not expected and raises concerns about the computations. The authors should provide more details and address this behavior explicitly.

Reviewer #3:

Remarks to the Author:

The manuscript from Du Prez and coworkers describes the synthesis and characterization of the earlier developed TAD systems as dynamic and force responsive linkers. The combination of mechanical and fluorescence characterization is a strength of the manuscript. The system is an elegant example of force activated chemistry, and should be published. However, there remain several aspects that need to be resolved.

1. Clarification of the impact on activation energies of the system (such as outlined by Ye et al. DOI: <https://doi.org/10.1039/C9MH01938C> or <https://doi.org/10.1021/acsmacrolett.6b00822>, which should be cited) or if it the case that simply enabling dynamic exchange to occur under load. The grinding data or fluorescence data on notched samples seem to suggest, along with the associated discussion, that the force is promoting bond severing. This is different to other examples in the literature, for instance the work of Sottos et al. which showed the activation energy increase with applied stress in their Spiropyran-Merocyanine system.
2. The grinding data is difficult to independently assess since presumably the effect of grinding is also to raise the surface area, which would accelerate dissolution. Discussion on how the authors could decouple these two factors would benefit their analysis? Certainly this supports the proposed work, but there could be several competing effects.
3. The irreversible vs dual vs regular TAD could be explained in additional detail. What is the origin of this distinctly different behavior? To the non expert, the structures in Fig 4c are almost identical so

further description of the origin of these differences in reactivity would help the reader understand how the design principles apply broadly.

4. Related to the above point, Further discussion of how the hysteresis is impacted by the bonding motif would be beneficial, perhaps comparing the energy of the loading vs dissipated in a more quantitative manner. This would identify potential changes in the material properties across the bonding motifs. This is because the data in Fig 4e do not vary all that much. Clearly effects beyond just the bonding of the units must be at play to have such significant energy dissipation even in the irreversible system.

5. One experiment that could be quite interesting is repeated stress relaxation, perhaps once at a low strain, comfortably in the small strain regime, followed by a stress relaxation in the medium. Or medium followed by High. This could really help quantify the effects highlighted by the authors in Fig 5c

6. The discussion section is less of a discussion against the broad knowledge in the literature and more of a conclusion. It would be better to either have discussion of results against the broader literature folded in to the remainder of the results section and have the last part more the conclusions and implications more broadly of the work.

Comments from the reviewer 1:

1. The manuscript by Du et al reports on the remarkable improvement of mechanical properties of crosslinked polyacrylate networks containing reversible covalent bond between TAD and phenylindole. The manuscript is clear in its message, the experiments are described clearly and support the conclusions. Interesting features are the intrinsic signalling of bond scission by spectroscopic means.

The manuscript is somewhat sloppy in its referencing (I found that many references do not correspond to the numbers in the reference list), and English usage is at places insufficient to understand the exact intention of the authors. For example in the caption to Fig. 2, the term 'forced direction' is not clear, and the word 'strengthening' in Fig 3b should probably be 'strengthening', but I am not sure. These weaknesses do not mean that the work should not be published, minor revision is sufficient.

Response: We appreciate the positive evaluation of our research. We updated the reference list in the revised manuscript and made it richer and more supportive regarding the state-of-the-art. Additionally, the English in the revised manuscript has been further polished.

2. I do however feel that the novelty of the work is insufficient for publication in Nature Communications. There are remarkable parallels with the work of Otsuka and coworkers (Sakai et al. Visualization and Quantitative Evaluation of Toughening Polymer Networks by a Sacrificial Dynamic Cross-Linker with Mechanochromic Properties. ACS Macro Lett. 2020, 9 (8), 1108–1113. <https://doi.org/10.1021/acsmacrolett.0c00321>.) Both studies use a reversible covalent bond to crosslink acrylates and demonstrate both toughening and visualization. Key features such as dissociation energy (124 vs 98 kJ/mol) are similar.

Because the chemistry of the reversible bond is different, and the reported toughening is stronger in the current manuscript publication of this work is certainly of interest, but does not have the novelty to merit publication in NC. I recommend publication in a polymer journal after revision that includes a comparison with the paper of Otsuka, and correction of the errors noted above (referencing and language).

Response: We regret that we have not explained our novelty clearly enough in our original manuscript. Compared with the reported research, the key innovation of our approach is multifold: **firstly, in our approach, TADs have a unique reaction selectivity with indole groups in dry-state polymer materials under ambient condition.** The TAD-indole chemistry adheres to the 'click' chemistry principles (*Nat. Chem.* **2014**, 6, 815) and offers selective covalent links that are quantitatively formed at

ambient temperature, and the TAD-indole adducts are known to be bench-stable. In contrast, the previously reported dynamic bonding system based on difluorenylsuccinonitriles (DFSN), which is referred to by the reviewer, relies on a homolytic dissociation into cyanofluorenyl radicals. Similar to most other reported radical systems, they are hence more susceptible to undergo irreversible reactions, for instance, with molecular oxygen, or other surrounding molecules (e.g. the polymer backbone). This would limit its practical use in the context of dynamic covalent bonding/re-bonding (*Mater. Chem. Front.* **2019**, *3*, 2270). It is inevitable for radical species to lose their activity when exposed to ambient condition for repeatedly long times to de-bond and re-bond. This could also implicate a decreasing number of crosslink points within the material. Even for relatively stable radicals, this would hamper the application under long-term force perturbation of the traditional toughened polymer materials.

Secondly, TAD-indole adducts, acting as crosslink points in dry-state covalently crosslinked polymers, enable materials to display reversible stress-responsiveness in real time, already at ambient temperature. In Otsuka's work, the difluorenylsuccinonitrile (DFSN) linker dissociates when subjected to a stretching force. However, the recovery of DFSN linker needs too much time to achieve self-reassociation in real time during stretching tests (the recovery of DFSN linker takes 30 min, as shown in the figure below, *ACS Macro Lett.* **2020**, *9*, 1108). In our work, the real-time cleavage and reconstruction of TAD-indole adducts has been confirmed by visual fluorescence mapping in Fig. 3e. When removing the external force, the fluorescence intensity recovers to the original state immediately, which implies the real-time dynamic behavior of TAD-indole adducts.

In order to further understand this real-time dynamic behavior, and to strengthen our revised manuscript, we designed an optical setup by combining a universal tester machine with a portable fluorescence spectrometer (Fig. 5a). This setup allowed us to quantitatively characterize the dynamic behavior of the reversible TAD-indole adducts during universal stretching tests. During the small stains ($\epsilon \leq 40\%$, cycle 1 and 2 in Fig. 5b), thickness-corrected normalized fluorescence intensity almost did not change, and the unbroken reversible C-N bonds act as conventional chemical crosslinks. At medium applied strains (ϵ , $\sim 60\%$, cycle 3 in Fig. 5b), the force-reversible C-N bonds in the network can be assumed to begin to break, thus causing the fluorescence intensity to sharply increase. When the stressed polymer materials are under high strain ($\epsilon \geq 80\%$, cycle 4 in Fig. 5b), the cleavage and reconstruction of the TAD-indole adducts reaches a dynamic equilibrium, revealing the same reaction rate for the force-reversible C-N de-bonding and re-bonding at high strain, and the fluorescence intensity originating from the recovery of TAD-indole adducts decreased rapidly in real time upon removing the stress.

The multiple breaking and reforming of the force-reversible C-N bond in TAD-indole adducts under force perturbation, endows dynamic covalent C-N crosslinked polymeric materials with

simultaneously enhancing mechanical strength and ductility, force-adaptability to dynamic environments, and network autonomy. This novelty in our approach resulted in the reported toughened effect, which entails an extraordinarily 2-5 fold increase in tensile toughness compared to Otsuka's work, increasing from 16.9 MJ m⁻³ to 34.7 MJ m⁻³ (CPMMA) and 74.1 MJ m⁻³ (CPMA). Such ambient force-reversible C-N covalent crosslinking platform could thus be considered as an important starting point for the design of longer-lasting toughened polymer materials.

ACS Macro Lett. 2020, 9, 1108-1113:

which revealed new bands at 509 and 542 nm (Figure S6), which were assigned to the cleavage of DFSN.²³ The intensity of the coloring after stretching increased with increasing DFSN content at the cross-linking points. The color faded after 30 min, probably due to recombination of the CF radicals. Therefore, for EPR measurements, the broken samples were maintained at a temperature sufficiently below the T_g to

need too much time to recovery in real time

Figure 4. (a) EPR spectra of CP_{DFMA-1} as a function of increasing strain. (b) Part of the typical stress-strain curve of CP_{DFMA-1} and the ratio of dissociated DFSN as a function of the strain.

only the cleavage of the central DFSN bond can be observed but no the recovery

In our work (Fig. 5ab):

In response of the reviewer's comment, the following description has been added to our revised manuscript, which discusses the parallels to the referred work, while also highlighting the particular challenges related to it.

“In addition, homolytic cleavage into relatively stable organic radicals showed promising reversible dissociation/association.^{19,20} However, radical recombination reactions are often susceptible to undergo irreversible reactions, for example with molecular oxygen, moisture, and other surrounding molecules, which is a concern in the context of dynamic covalent chemistry.²⁰ In particular when exposed to ambient conditions for a long time, radical species are expected to lose their activity to de-bond and re-bond. Nonetheless, Otsuka et al.²¹ developed a notable example based on a difluorenylsuccinonitrile (DFSN) linker whose central C-C bond is readily cleaved under mechanical stress to generate a relatively stable radical species. However, the delayed recovery of the DFSN linkers did not allow for their self-reassociation in real time.”

“Measurements for load, time and fluorescence intensity were captured simultaneously during cyclic tensile testing using a combination of an optical and mechanical setup. A photo and schematic of the setup are shown in Fig. 5a. The force-reversible behavior of TAD-indole adducts during the stretching tests is manifested as the thickness-corrected normalized fluorescence intensity value ($I_{fl, norm}$, details in Methods). At small applied strains ($\epsilon \leq 40\%$, Fig. 5d), $I_{fl, norm}$ almost did not change (cycle 1 and 2 in Fig. 5b), and the unbroken reversible C-N bonds act as conventional chemical crosslinks. The resulting macroscopic deformation corresponds to chain conformation changes, and the entropy of the system is reduced. The unbroken bonds preserve shape-memory and drives the network entropically back to the unstretched state after the force is released (Supplementary Fig. 26a,b). At medium applied strains (ϵ , $\sim 60\%$), the force-reversible C-N bonds in the network can be assumed to begin to break (Fig. 5d), thus causing $I_{fl, norm}$ to sharply increase (cycle 3 in Fig. 5b) with a large deformation recovery of the network (Supplementary Fig. 26c). Finally, at larger applied strains ($\epsilon \geq 80\%$), the breakage of reversible C-N bonds occurs during elongation while their reformation can happen at newly accessible sites (Fig. 5d). Thus, no entropy is gained during the macroscopic deformation leading to network topological rearrangement. In essence, the network is in this case permanently deformed (non-recoverable) from the standpoint of thermodynamics, and the residual strain would be observed (illustrated in Fig. 4d and Supplementary Fig. 26d,e).⁴⁷ In addition, the cleavage and reconstruction of TAD-indole adducts reaches a dynamic equilibrium (fixed $I_{fl, norm}$), revealing the same reaction rate of force-reversible C-N de-bonding and re-bonding at high strain (cycle 4 in Fig. 5b). During retraction, the fluorescence intensity originating from the recovery of TAD-indole adducts decreased rapidly in real time with the stress being removing (cycle 3 and 4). The modulus of the polymer network after TAD-indole adducts re-association maintained around 40 MPa (cycle 3 and 4). Breaking and reforming kinetics of the dynamic bonds in the network will significantly affect the mechanical properties of the polymer materials. A repeated stress relaxation curve corroborates the fast kinetics of the force-reversible C-N bond in the network ($\epsilon > 60\%$, Fig. 5c), which is essential for the design of tough, high performance polymer materials.”

Comments from the reviewer 2:

1. This work addresses a quite interesting topic and is potentially well suited for publication in NCOMMS. The central hypothesis is that introducing a room-temperature forming dynamic covalent bond as cross-linker provides a site of energy dissipation that can also rapidly re-form at room temperature. This could potentially lead to longer-life materials.

The presented evidence is qualitatively supportive of/consistent with the hypothesis, but additional and important control experiments are needed. Changes in bulk properties can

arise from very small structural perturbations, and the addition of the TAD cross-linker itself could act as a plasticizer and/or anti-plasticizer that complicates the origins of the changes in mechanical properties. It is possible that while some bond breaking and bond re-forming are occurring, the changes in mechanical properties are due to other effects altogether. So the data in Fig 4a and 4b is difficult to interpret.

Response: We thank the reviewer for the very positive evaluation of our work and we appreciate the good suggestion. Additional control experiments have thus been added in our revised manuscript. To evaluate the effect of TAD crosslinker itself as a plasticizer and/or anti-plasticizer, a small amount of Ph-TAD (5 mol%), which is monofunctionality, was introduced into the LPMA or LPMMA polymer materials (LPMA* and LPMMA*). Ph-TAD has a same chemical structure with the corresponding MDI-TAD structure (shown in the figure below). So, Ph-TAD is not a crosslinker TAD, but a modification agent. The stress-strain curves for this control experiment are shown in Supplementary Fig. 22. It indicates that the introduction of a small amount of Ph-TAD (5 mol%) almost has no effect on the mechanical properties. Hence, the effect of the addition of a small amount of TAD crosslinker on the mechanical properties can be neglected.

The corresponding description has been added in our revised manuscript: *“To evaluate the effect of the addition of the TAD crosslinker on the polymeric mechanical properties, a foreseen control experiment was performed by the addition of Ph-TAD, the monofunctional TAD modification agent that has a similar chemical structure as the crosslink points (Supplementary Fig. 6). Thus, a small amount of Ph-TAD (5 mol%), was introduced into the LPMA or LPMMA polymer materials (LPMA* and LPMMA*). Uniaxial tensile tests showed that the introduction of a small amount of Ph-TAD has no significant effect on the mechanical properties (Supplementary Fig. 22). Therefore, the effect of the addition of a small amount of TAD crosslinker itself, acting as a plasticizer or anti-plasticizer, on the mechanical properties was not considered further in this study.”*

Supplementary Figure 22. Stress as a function of strain of **a** LPMA and LPMA* ($\dot{\epsilon} = 20 \text{ mm min}^{-1}$) and **b** LPMMA and LPMMA* ($\dot{\epsilon} = 2 \text{ mm min}^{-1}$) films. LPMA* and LPMMA* films have the same amount of Ph-TAD reactive with indole (5 mol%) in the polymer network.

2. The data in Fig 4d/e is more compelling, but the difference in high strain behavior can be greatly influenced by small differences in cross-linking density. I am especially struck by the fact that the green curve eventually stays at higher modulus than the blue curve, which is counter to the picture presented of easier bond breaking in green than blue. One explanation is that the rate of cross-link formation in the irreversible adducts is different from the rate of reversible adducts, leading to "patches" of one and the other. If this is the case, it would also mean that the relative diffusion/reactivity in the two systems is different in forming the thermoset networks, and this could contribute even to the behavior of red vs. green, etc...

Response: Although the TAD-indole reaction is instantaneous and hence no significant differences in terms of reactivity are expected between the reversible and irreversible indole reaction partner that have been selected in this work, we further carefully explored the reason for the observed difference in high strain behavior. We found that the glass transition temperatures (T_g s) of du-CPMA, ir-CPMA and CPMA are 24 °C, 28 °C and 18 °C, respectively (see the figure below). In cyclic stretching tests (Fig 4d/e in the original manuscript), the testing temperature was 25 °C. In other words, ir-CPMA will behave more as a glassy and CPMA more as a rubbery polymer. Compared to CPMA, ir-CPMA polymer chain segments have less mobility, resulting in a decreased modulus in high strain. On the contrary, CPMA as a rubbery polymer exhibits molecular chain orientation in high strain, resulting in the mechanical property enhancement.

Supplementary Fig. 25 has been added:

Supplementary Fig. 10, CPMA film:

Supplementary Figure 25. Differential scanning calorimetry analysis (DSC) of du-CPMA and ir-CPMA films.

To exclude the effect of mobility for du-CPMA, ir-CPMA and CPMA polymers, we retested these three different crosslinked polymers at 40 °C to ensure they are all in the rubbery state. Additionally, in our revised manuscript, the maximum strain increased from 100% to 150 % in order to effectively present the contribution of the dynamic covalent bonds to energy dissipation. The cyclic stretching tests

at a strain of 150 % are displayed in Fig. 4d, and the CPMA polymer network is shown to exhibit the highest energy dissipation compared to du-CPMA and ir-CPMA.

The corresponding description has been added in our revised manuscript: “*The cyclic stretching tests were performed at 40 °C in order to maintain the same physical state for all materials (rubber, Supplementary Fig. 25).*”

“*Increased amounts of reversible TAD-indole adducts were conducive to reduce molecular internal stress due to the continuous reversible behavior embedded by the C-N bond of the TAD-indole adducts, thus resulting in the lowest strain recovery ($\epsilon_{recovery} = 58\%$) for CPMA compared to that of du-CPMA ($\epsilon_{recovery} = 114\%$) and ir-CPMA ($\epsilon_{recovery} = 148\%$).*”

Fig. 4 Uniaxial tensile and recovery. Stress as a function of strain of **a** CPMA ($\dot{\epsilon} = 20 \text{ mm min}^{-1}$) and **b** CPMMA ($\dot{\epsilon} = 2 \text{ mm min}^{-1}$) films. **c** Structural representation of irreversible crosslinked PMA (ir-CPMA) and dual crosslinked CPMA (du-CPMA) containing a 1:1 ratio of reversible and irreversible crosslinks. **d** One cyclic stretching of different crosslinked films at 40 °C. **e** The mechanical internal friction dissipation at different strains.

3. Along those lines, one worries that the comparison of Fig. 4e is potentially misleading. Is the same trend in energy dissipation observed at other strains (e.g., 20%, 40%, 60%, etc...)?

Response: Additional cyclic stretching tests at other strains (i.e., 20%, 40%, 60%, 80% and 100%) have been performed. CPMA exhibited more efficient energy dissipation after a strain of > 60% compared to du-CPMA and ir-CPMA, the latter showing the lowest energy dissipation. At low strain (< 40%), the external stress is too low to break the C-N bonds in the TAD-indole adducts. Therefore, CPMA, ir-CPMA and du-CPMA films exhibited a similar energy dissipation (within error) after one cyclic stretching. The data statistics of internal dissipation is displayed in Fig. 4e, with the cyclic stretching experiments depicted in Supplementary Fig. 26 (included here below).

The corresponding description has been added in our revised manuscript: “*To characterize potential changes in the mechanical properties across different bonding motifs, a quantitative assessment by means of equation (1) was used to reveal the relationship between the energy of loading and dissipation in the hysteresis:*

$$\psi = \frac{U_{dis}}{U_{loading}} \times 100\% \quad (1)$$

where ψ , $U_{loading}$ and U_{dis} refers to the mechanical internal friction dissipation and the energy of loading and dissipated energy, respectively. We first tested one cyclic stretching of three crosslinked polymer films at different strains (20%, 40%, 60%, 80%, 100% and 150 %, Supplementary Fig. 26 and Fig. 4d). At small applied strains ($\epsilon \leq 40$ %), three covalent crosslinked polymer samples had similar U_{dis} , $U_{loading}$ and ψ values (Fig. 4e), indicating that the external force at low strain is too low to break the reversible C-N bonds, making these to act as fixed crosslink points. At higher applied strains, i.e., $\epsilon \geq 60$ %, continuous dissociation and reconstruction of the C-N bonds in the TAD-indole adducts resulted in a large increase of U_{dis} and ψ for the CPMA samples, as compared with the ir-CPMA and du-CPMA samples (Fig. 4e). This implies that the force-reversible C-N bonds can efficiently dissipate the energy caused by external forces.”

Supplementary Figure 26. One cyclic stretching of different crosslinked films at 40 °C under different stretching strain of **a** 20%, **b** 40%, **c** 60, **d** 80%, **e** 100%

4. The optical data need to be shown in greater detail. There is no "zero" absorbance on the plots, so it is not clear how large the change in intensity relative to final intensity is. In particular, one wonders about the potential for photobleaching (not addressed).

Response: We entirely agree with the Reviewer's suggestion. In our revised manuscript, we revised all absorbance and fluorescence plots, e.g., Fig. 3b.

Additionally, we monitored the 460 nm fluorescence intensity changes for the LPMMA film at 365 nm excitation when irradiated for one hour (Supplementary Fig. 19). The LPMMA film showed excellent photostability. The corresponding description has been added in our revised manuscript: "*The LPMMA films showed excellent photostability under 365 nm when irradiated for one hour (Supplementary Fig. 19).*"

Supplementary Figure 19. The 460 nm fluorescence intensity changes for LPMMA film at 365nm excitation light irradiation for one hour.

5. And what fraction of the "reacted" TADs are actually forming mechanically active cross-links. How does the modulus of the network after this type of self-healing compare to the initial modulus?

Response: In our revised manuscript, we added FTIR data of MDI-TAD in Supplementary Fig. 8. The absence of the -N=N- characteristic peak in FTIR from the 'unreacted' TAD, evidenced quantitative (i.e., full TAD conversion) crosslinking of the CPMA and CPMMA films. We therefore assume all TAD-indole crosslinks can be mechanically activated. We further undertook additional experiments aimed to accurately characterize the fraction of crosslinks that participate in the mechanically activated self-healing. For this, we attempted to monitor relative indole content changes using fluorescence intensity during stretching experiments (cf. Fig. 5b), making use of a customised optical setup that combines a universal tester machine with a portable fluorescence spectrometer (Fig. 5a). During the small stains ($\epsilon \leq 40\%$, cycle 1 and 2 in Fig. 5b), thickness-corrected normalized fluorescence intensity almost did not change, and the unbroken reversible C-N bonds act as conventional chemical crosslinks. At medium applied strains ($\epsilon, \sim 60\%$, cycle 3 in Fig. 5b), the force-reversible C-N bonds in the network can be assumed to begin to break, thus causing the fluorescence intensity to sharply increase. When the

stressed polymer materials are under high strain ($\epsilon \geq 80\%$, cycle 4 in Fig. 5b), the cleavage and reconstruction of the TAD-indole adducts reaches a dynamic equilibrium, revealing the same reaction rate for the force-reversible C-N de-bonding and re-bonding at high strain, and the fluorescence intensity originating from the recovery of TAD-indole adducts decreased rapidly in real time upon removing the stress.

In cyclic tensile test (Fig. 5b), the modulus of the CPMA network in cycle 3 is almost the same compared to that of cycle 4.

The corresponding description has been added in our revised manuscript: “*High TAD conversion in indole-based polymer networks was confirmed by Fourier-transform infrared spectroscopy (Supplementary Fig. 8).*”

“*The modulus of the polymer network after TAD-indole adducts reassociation maintained around 40 MPa (cycle 3 and 4).*”

Supplementary Figure 8. Fourier transform infrared spectroscopy of **a** PMA, LPMA, CPMA, MDI-TAD and **b** PMMA, LPMMA, CPMMA, MDI-TAD.

6. Control experiments with HDEO and no grinding should be performed, as well as grinding of linear polymers with TAD-indole side chain adducts (or small molecule adducts within the cross-linked material) to rule out heat induced dissociation or the presence of trapped, small amounts of initially unreacted TAD.

Response: In order to rule out heat induced dissociation or the presence of trapped small amounts of initially unreacted TAD, the suggested additional reference experiments have been carried out. Specifically, we synthesized a linear PMMA with PhTAD-indole side chain adducts (LPMMA-TAD), using a monofunctional TAD modification agent instead of the bifunctional TAD crosslinker. The resulting LPMMA-TAD was soaked into a 1 M ethanol solution of HDEO (Fig. 2a) to form the mixture containing LPMMA-TAD and HDEO (LPMMA-TAD+HDEO). It was found that, after grinding the dry LPMMA-TAD-HDEO sample, the low molecular weight translinked product (TAD-HDEO) was

obtained, as verified by ^1H NMR (Fig. 2b). For comparison, there is no characteristic peak of the TAD-HDEO observed in the ^1H NMR of the unground, dry LPMMA-TAD-HDEO sample (Fig. 2b). Thus, we can conclude that the nascent TAD moiety, released during the applied grinding force, is believed to be consumed in the presence of HDEO.

The corresponding description has been added in our revised manuscript: “*Since TAD-indole adducts can be ‘transclicked’ upon heating in the presence of a conjugated diene, i.e., 2,4-hexadien-1-ol (HDEO), to generate an irreversible TAD-HDEO adduct (Fig. 2a),²⁴ the LPMMA film with TAD-indole side chain adducts was cut by scissors and soaked into a 1 M ethanol solution of HDEO. The LPMMA film containing HDEO was received after drying the swollen network at 40 °C for 12 h to form the mixture containing LPMMA and HDEO. It was found that, after grinding the mixture, TAD-HDEO adducts were obtained as verified by ^1H NMR spectroscopy of the dissolved ground powder in DMSO- d_6 (Fig. 2b). For comparison, there is no characteristic peak of the TAD-HDEO in ^1H NMR for the unground reference sample (Fig. 2b). Thus, the nascent TAD moiety released during the applied grinding force is believed to be consumed in the presence of HDEO.*”

Fig. 2 Verified force-reversible C-N bond breaking in the dry-state polymer network. **a** Transclick reaction of TAD-indole side chain in LPMMA to an irreversible TAD-HDEO adduct, triggered by the external force at ambient temperature. **b** ^1H NMR spectrum of a small molecule HDEO (blue), the TAD-HDEO reference adduct (black), the mixture of HDEO and LPMMA with TAD-indole side chains (red) and the ground mixture (green) dissolved in DMSO- d_6 . **c** Free energy profiles (kJ/mol) for TAD-indole adduct, open zwitterionic intermediate (OI) and transition state (OI-TS). **d** Theoretically calculated restored force, free energy and potential energy of dissociation of TAD-indole adducts as a function of C-N bond length at 1 atm, 298.15 K. **e** TAD-indole adduct with yellow arrows along the C-N bond representing the loading direction of the C and N atom when external force is applied in the quantum-chemical calculations.

7. On page 6, the authors state that the "larger stabilization of ground state" makes the system

more reactive, but this seems backwards. Ground state stabilization leads to lower reactivity.

Response: Indeed, we apologize for this misleading explanation in our original manuscript. It is evident that the ground state of TAD-indole adducts is the most stable state compared to the intermediate state, transition state and the pre-reactive TAD and indole complex, which is consistent with the highly exergonic TAD-indole click reaction at ambient temperature (see the figure below, taken from *Nature Chemistry*, 2014, 6, 815-821). In our work, we simulated the polymer-relative click chemical structure using aromatic TAD (Ph-TAD) and 2-penylindole to calculate the change in Gibbs free energy of the adducts, open zwitterionic intermediate state and transition state in Fig. 2d. According to the simulations, the external force can be introduced to activate TAD-indole adducts (G_{adducts} increases), together with the decrease of dissociation activation energy and ΔG .

The corresponding description has been added in our revised manuscript: “*The ΔG dependence of the constrained TAD-indole adducts on the force resulted from the differences in stabilization of the ground states and the transition state (Fig. 2c, Supplementary Table 3). As expected, with the increasing force, the de-stabilizing TAD-indole adducts lead to the reaction equilibrium to favor dissociation, with a concomitant decrease in the ΔG (Fig. 2d).*”¹¹

8. The discussion of "force-activated bonds" is a bit too general, as even conventional covalent bonds are force-activated to break (as noted elsewhere in the paper). This terms should be better defined in the context of the current paper.

Response: We entirely agree with the Reviewer's suggestion. Hence, we re-defined "force-activated bonds" as "weak force-activated covalent bonds" to distinguish it from conventional covalent bonds. And we re-defined "force-activated reversible C-N bonds" as "force-reversible C-N bonds" in our

revised manuscript.

9. References 13 and 14 are used to support the statement that "Currently, a series of force-activated covalent bonds have been introduced as crosslinking points into covalent polymer networks to increase simultaneously the mechanical strength and ductility of a polymeric material." But I do not see discussion of force-activated bonds in this way in what are very general reviews of dynamic covalent bonds in polymers. The authors might be better served by pointing to primary literature here.

Response: Thanks to the reviewer for the good suggestion. In our revised manuscript, we have cited more relevant literature examples to support our discussion on force-activated covalent bonds as crosslinking points to increase polymeric mechanical properties.

13. Qi, Q. et al. Force-induced near-infrared chromism of mechanophore-linked polymers. *J. Am. Chem. Soc.* **143**, 17337-17343 (2021).

14. Lou, J., Friedowitz, S., Will, K., Qin, J. & Xia, Y. Predictably engineering the viscoelastic behavior of dynamic hydrogels via correlation with molecular parameters. *Adv. Mater.* **33**, 2104460 (2021).

10. The computed force profile in Fig 2e is very strange and uncharacteristic of most COGEF type calculations. The computational details are quite short and more should be provided. But the downward curvature at high forces is not expected and raises concerns about the computations. The authors should provide more details and address this behavior explicitly.

Response: We appreciate this good suggestion. In our work, we used the enforced geometry optimization (EGO) model to study the maximum force of the C-N bonds of TAD-indole adducts. Similar downward curvature at high forces have been reported in Beyer's work (*J. Chem. Phys.* **112**, 7307-7312 (2000)), as shown in figure below extracted from this literature example. When the molecule is further stretched, it becomes energetically favorable to the expense of the one that breaks, in this case C-N. The demanding energy of further activating the TAD-indole adducts is decreasing gradually, and the slope of potential energy and the distance between two atoms is certainly decreasing, which means the force of C-N bonds is decreasing. The changing trend is the same as in Beyer's work who have studied the maximum force of different covalent bonds, incl. C-N, C-C, C-O, Si-C, in model molecules.

In response to the reviewer's concern, we modified Fig. 2d in which additional potential energy data have been introduced, making it more clear to correspond to the energy change with regard to the force. More computational details have been added in the section of quantum-chemical calculations: *“The enforced geometry optimization (EGO) model was used as the computational method that considers force explicitly under isotensional stretching to study the maximum force of C-N bonds of TAD-indole adducts.²⁹ The mechanical stretching of a molecule was simulated by relaxed potential energy surface scans. The distance r between the two atoms (C and N atom) on which the force is applied is taken as the reaction coordinate. Starting from the equilibrium geometry, $r(\text{C-N})$ is elongated in fixed small steps (0.05 \AA), and with the distance r fixed. An otherwise fully relaxed geometry optimization is performed at each step, giving the potential as a function of r . The maximum slope of potential determines the maximum force of the C-N bonds.”*

29. Beyer, M. K. The mechanical strength of a covalent bond calculated by density functional theory. *J. Chem. Phys.* **112**, 7307-7312 (2000).

Fig. 2 Verified force-reversible C-N bond breaking in the dry-state polymer network. **a** Transclick reaction of TAD-indole side chain in LPMMA to an irreversible TAD-HDEO adduct, triggered by the external force at ambient temperature. **b** ^1H NMR spectrum of a small molecule HDEO (blue), the TAD-HDEO reference adduct (black), the mixture of HDEO and LPMMA with TAD-indole side chains (red) and the ground mixture (green) dissolved in $\text{DMSO-}d_6$. **c** Free energy profiles (kJ/mol) for TAD-indole adduct, open zwitterionic intermediate (OI) and transition state (OI-TS). **d** Theoretically calculated restored force, free energy and potential energy of dissociation of TAD-indole adducts as a function of C-N bond length at 1 atm, 298.15 K. **e** TAD-indole adduct with yellow arrows along the C-N bond representing the loading direction of the C and N atom when external force is applied in the quantum-chemical calculations.

Comments from the reviewer 3:

1. The manuscript from Du Prez and coworkers describes the synthesis and characterization of the earlier developed TAD systems as dynamic and force responsive linkers. The combination of mechanical and fluorescence characterization is a strength of the manuscript. The system is an elegant example of force activated chemistry, and should be published. However, there remain several aspects that need to be resolved.

Clarification of the impact on activation energies of the system (such as outlined by Ye et al. DOI:<https://doi.org/10.1039/C9MH01938C> or <https://doi.org/10.1021/acsmacrolett.6b00822>, which should be cited) or if it the case that simply enabling dynamic exchange to occur under load. The grinding data or fluorescence data on notched samples seem to suggest, along with the associated discussion, that the force is promoting bond severing. This is different to other examples in the literature, for instance the work of Sottos et al. which showed the activation energy increase with applied stress in their Spiropyran-Merocyanine system.

Response: We appreciate the very positive evaluation of our work. In our revised manuscript, we have cited the abovementioned literature. We added the activation energy changes of the dissociation reaction in Supplementary Fig. 14 and the corresponding description has been added in our revised manuscript:

“Applied external force on the polymers is capable of activating internal C-N bonds of the TAD-indole adducts and reducing the apparent activation barrier for the dissociation reaction, implying that free states of TAD and indole groups are more favorable with higher stress (Supplementary Fig. 11). This result is consistent with prior reports that applied force affects the reversible reaction tendency by decreasing the activation energy barrier for the dissociation reaction.^{32,33}”

Supplementary Figure 11. Effect of stress on activation energy before the maximum force applied on TAD-indole adducts.

We regret that we have not described our view clearly in the original manuscript. In fact, the change tendency of activation energy is consistent with that of Sottos’ work (*ACS Macro Lett.* **5**, 1312-1316 (2016)) as can be observed in the figure below. The author also described that the applied force will lower the activation energy of the dissociation reaction.

2. The grinding data is difficult to independently assess since presumably the effect of grinding is also to raise the surface area, which would accelerate dissolution. Discussion on how the authors could decouple these two factors would benefit their analysis? Certainly this supports the proposed work, but there could be several competing effects.

Response: According to the suggestion from **Reviewer 2** and **Reviewer 3**, in order to rule out the effect of increasing surface area after grinding in our revised manuscript, we carried out control experiments on a linear polymer. Thus, we synthesized a linear PMMA with PhTAD-indole side chain adducts (LPMMA-TAD), and then LPMMA-TAD was soaked into a 1 M ethanol solution of HDEO (Fig. 2a) to form the mixture containing LPMMA-TAD and HDEO (LPMMA-TAD-HDEO). It was found that, after grinding the dry LPMMA-TAD-HDEO sample, the translicked product (TAD-HDEO) was obtained as verified by ^1H NMR (Fig. 2b). For comparison, there is no characteristic peak of the TAD-HDEO in ^1H NMR for ungrounded dry LPMMA-TAD-HDEO sample (Fig. 2b). Thus, we can conclude that the nascent TAD moiety released during the applied grinding force is believed to be consumed in the presence of HDEO.

The corresponding description has been added in our revised manuscript: “*The LPMMA film containing HDEO was received after drying the swollen network at 40 °C for 12 h to form the mixture containing LPMMA and HDEO. It was found that, after grinding the mixture, TAD-HDEO adducts were obtained as verified by ^1H NMR spectroscopy of the dissolved ground powder in DMSO- d_6 (Fig. 2b). For comparison, there is no characteristic peak of the TAD-HDEO in ^1H NMR for the unground reference sample (Fig. 2b). Thus, the nascent TAD moiety released during the applied grinding force is believed to be consumed in the presence of HDEO.*”

Fig. 2 Verified force-reversible C-N bond breaking in the dry-state polymer network. **a** Translick reaction of TAD-indole side chain in LPMMA to an irreversible TAD-HDEO adduct, triggered by the external force at ambient temperature. **b** ^1H NMR spectrum of a small molecule HDEO (blue), the TAD-HDEO reference adduct (black), the mixture of HDEO and LPMMA with TAD-indole side chains (red)

and the ground mixture (green) dissolved in DMSO-*d*₆. **c** Free energy profiles (kJ/mol) for TAD-indole adduct, open zwitterionic intermediate (OI) and transition state (OI-TS). **d** Theoretically calculated restored force, free energy and potential energy of dissociation of TAD-indole adducts as a function of C-N bond length at 1 atm, 298.15 K. **e** TAD-indole adduct with yellow arrows along the C-N bond representing the loading direction of the C and N atom when external force is applied in the quantum-chemical calculations.

3. The irreversible vs dual vs regular TAD could be explained in additional detail. What is the origin of this distinctly different behavior? To the non expert, the structures in Fig 4c are almost identical so further description of the origin of these differences in reactivity would help the reader understand how the design principles apply broadly.

Response: We understand that this could be difficult to comprehend without prior knowledge about reversible TAD chemistry. The reversible vs irreversible indole linkage with TAD highly depends on the substituents in the indole C2/C3-position. If there is no substituent on the indole C2-position, the TAD is thermodynamically favored to add in the indole C2-position. Even when initial reaction takes place on the indole C3-position (which is the most electron rich), rearrangement to the C2-TAD adduct will take place, so that the aromaticity of the indole will not be lost in the resulting addition product. This preservation of the aromatic ring makes such a TAD addition, in practice, irreversible. This can also be rationalized from the lack of a driving force for TAD-indole dissociation, since no aromaticity is regained (in contrast to the C3-TAD adduct). We have included additional information that is relevant to the reversible vs. irreversible TAD-indole reaction in the manuscript, and referred to a PhD dissertation (doi:10.5445/IR/1000100603) where the reader can find more details about this.

The corresponding description has been added in our revised manuscript: “*The reversible vs. irreversible linkage of TADs with indoles highly depends on the substituents in the indole C2 and C3-position. When no substituent on the indole C2-position is present, as is the case for NIAM, TAD-addition will result in the thermodynamically favored adduct in the C2-position (Supplementary Fig. 23). Consequently, the retention of aromaticity in the C2-indole adduct renders the TAD addition irreversible in practice.*”⁴⁶”

4. Related to the above point, Further discussion of how the hysteresis is impacted by the bonding motif would be beneficial, perhaps comparing the energy of the loading vs dissipated in a more quantitative manner. This would identify potential changes in the material properties across the bonding motifs. This is because the data in Fig 4e do not vary all that much. Clearly effects beyond just the bonding of the units must be at play to have such significant energy dissipation even in the irreversible system.

Response: We entirely agree with the Reviewer’s suggestion. As explained earlier, we used a more quantitative analysis, using the mechanical internal friction dissipation (ψ) (the ratio of dissipated (U_{dis})

and loading energy (U_{loading}), to identify potential changes in the material properties across the bonding motifs. Additionally, according to the suggestion from **Reviewer 2** and **Reviewer 3**, to exclude the effect of mobility for du-CPMA, ir-CPMA and CPMA polymers, we retested these three different crosslinked polymers at 40 °C to ensure that they are all in the rubbery state. Additionally, the maximum strain increased from 100% to 150 % in order to effectively present the contribution of the dynamic covalent bonds to the energy dissipation. Cyclic stretching tests at other strains (20%, 40%, 60%, 80%, 100%) have been performed and are presented in Supplementary Fig. 26. CPMA exhibited the most efficient energy dissipation at strains > 60% compared to ir-CPMA and du-CPMA. At low strain (< 40%), external stress is too low to break the C-N bonds in TAD-indole adducts, therefore, CPMA, ir-CPMA and du-CPMA films exhibited similar energy dissipation (within error) after one cyclic stretching. Consequently, the data statistics of internal dissipation displayed in Fig. 4e have been changed.

The corresponding description has been added in our revised manuscript: “*To characterize potential changes in the mechanical properties across different bonding motifs, a quantitative assessment by means of equation (1) was used to reveal the relationship between the energy of loading and dissipation in the hysteresis:*

$$\psi = \frac{U_{\text{dis}}}{U_{\text{loading}}} \times 100\% \quad (1)$$

where ψ , U_{loading} and U_{dis} refers to the mechanical internal friction dissipation and the energy of loading and dissipated energy, respectively. We first tested one cyclic stretching of three crosslinked polymer films at different strains (20%, 40%, 60%, 80%, 100% and 150 %, Supplementary Fig. 26 and Fig. 4d). At small applied strains ($\epsilon \leq 40$ %), three covalent crosslinked polymer samples had similar U_{dis} , U_{loading} and ψ values (Fig. 4e), indicating that the external force at low stain is too low to break the reversible C-N bonds, making these to act as fixed crosslink points. At higher applied strains, i.e., $\epsilon \geq 60$ %, continuous dissociation and reconstruction of the C-N bonds in the TAD-indole adducts resulted in a large increase of U_{dis} and ψ for the CPMA samples, as compared with the ir-CPMA and du-CPMA samples (Fig. 4e). This implies that the force-reversible C-N bonds can efficiently dissipate the energy caused by external forces.”

Fig. 4 Uniaxial tensile and recovery. Stress as a function of strain of **a** CPMA ($\dot{\epsilon} = 20 \text{ mm min}^{-1}$) and **b** CPMMA ($\dot{\epsilon} = 2 \text{ mm min}^{-1}$) films. **c** Structural representation of irreversible crosslinked PMA (ir-CPMA), and dual crosslinked CPMA (du-CPMA) containing a 1:1 ratio of reversible and irreversible crosslinks. **d** One cyclic stretching of different crosslinked films at 40 °C. **e** The mechanical internal friction dissipation of the three networks at different strains.

5. One experiment that could be quite interesting is repeated stress relaxation, perhaps once at a low strain, comfortably in the small strain regime, followed by a stress relaxation in the medium. Or medium followed by High. This could really help quantify the effects highlighted by the authors in Fig 5c.

Response: We performed the repeated stress relaxation tests at $\epsilon = 40\%$, 60% and 80% , respectively. The results in Fig. 5c indicated the fast kinetics of the force-reversible bond exchanges when the C-N bonds in the network begin to break ($\epsilon > 60\%$).

The corresponding description has been added in our revised manuscript: “A repeated stress relaxation curve corroborates the fast kinetics of the force-reversible C-N bond in the network ($\epsilon > 60\%$, Fig. 5c), which is essential for the design of tough, high performance polymer materials.”

Fig. 5 The recovery mechanism and kinetics of the reversible C-N bonds. **a** Design of the customized setup, combining the use of a universal testing machine and portable fluorescence spectrometer, $\lambda_{exc} = 365$ nm. **b** Simultaneously obtained stress and thickness-corrected normalized fluorescence intensity as a function of strain. **c** The repeated stress relaxation of CPMA at $\epsilon = 40\%$, 60% and 80% , respectively. Relaxation time is 10 min for every stage. **d** Schematic representation of proposed bond rearrangements for the strain recovery.

6. The discussion section is less of a discussion against the broad knowledge in the literature and more of a conclusion. It would be better to either have discussion of results against the broader literature folded in to the remainder of the results section and have the last part more the conclusions and implications more broadly of the work.

Response: We agree with the Reviewer's suggestion and have included the discussion part within the results section of the manuscript, so that the last paragraph presents a conclusion of the work.

Reviewers' Comments:

Reviewer #3:

Remarks to the Author:

The reviewer is satisfied with the responses to the reviewer's concerns.

Reviewer #4:

Remarks to the Author:

Overall, the authors have addressed Reviewer 2's comments well. I have some further question about response 6, where grinding on the linear polymer with TAD shows transclick. This behavior is distinct from polymer mechanochemistry, where polymer chains are attached to both sides of a mechanophore. Do the authors attribute this observation to thermal activation or mechanical activation? In addition, in their response 5, the authors "assume all TAD-indole crosslinks can be mechanically activated" which is not what I would expect. Force is not evenly distributed in a polymer network and only the stretched subchains would bear force for mechanical activation. Other than these, the paper is in good shape, and I recommend its publication in Nature Communications.

Comments from the reviewer 4:

1. Overall, the authors have addressed Reviewer 2's comments well. I have some further question about response 6, where grinding on the linear polymer with TAD shows transclick. This behavior is distinct from polymer mechanochemistry, where polymer chains are attached to both sides of a mechanophore. Do the authors attribute this observation to thermal activation or mechanical activation?

Response: We appreciate the positive evaluation of our research. Grinding test would apply external force to polymer or small molecule [ref. 1], both of which is feasible. Mechanochemistry in side chain can be stressed during grinding. ¹H NMR spectrum of the mixture of HDEO and LPMMA with TAD-indole side chains before and after grinding is to rule out heat induced dissociation, as the reviewer 2 suggested. To avoid overheating caused by friction, we refer to the reported grinding method [ref. 2] to maintain ambient temperature (30 minutes of grinding with intervals of 10 minutes by using agate mortar). And dissociation of TAD-indole adducts requires high temperature (120 °C) [ref. 3]. In this method, the temperature of agate mortar increasing to 120 °C is impossible. Therefore, the transclick observation can attribute to mechanical activation. We added more discussion in revised manuscript: *“The grinding time was 30 minutes with intervals of 10 minutes to avoid overheating caused by friction.”*; *“Thus, the nascent TAD moiety due to mechanical activation during the applied grinding force is believed to be consumed in the presence of HDEO.”*

[ref. 1] *ACS Macro Lett.* **7**, 1359-1363 (2018).

[ref. 2] *J. Am. Chem. Soc.* **143**, 17744-17750 (2021).

[ref. 3] *Nat. Chem.* **6**, 815-821 (2014).

2. In addition, in their response 5, the authors "assume all TAD-indole crosslinks can be mechanically activated" which is not what I would expect. Force is not evenly distributed in a polymer network and only the stretched subchains would bear force for mechanical activation.

Response: We originally meant that all TAD-indole crosslinks have the potential of force-response. The fact as the reviewer said, force is not evenly distributed and the stretched subchains would bear force for mechanical activation.